# The Missing Indicator Approach for Accelerated Failure Time Model with Covariates Subject to Limits of Detection

**Norah Alyabs** [1,2,*] and **Sy Han Chiou** [1]

1    Department of Mathematical Sciences, The University of Texas at Dallas, Richardson, TX 75080, USA;
     schiou@utdallas.edu
2    College of Sciences and Theoretical Studies, Saudi Electronic University, Riyadh 13316, Saudi Arabia
*    Correspondence: norah.alyabs@utdallas.edu or n.alyabs@seu.edu.sa

**Abstract:** The limit of detection (LOD) is commonly encountered in observational studies when one or more covariate values fall outside the measuring ranges. Although the complete-case (CC) approach is widely employed in the presence of missing values, it could result in biased estimations or even become inapplicable in small sample studies. On the other hand, approaches such as the missing indicator (MDI) approach are attractive alternatives as they preserve sample sizes. This paper compares the effectiveness of different alternatives to the CC approach under different LOD settings with a survival outcome. These alternatives include substitution methods, multiple imputation (MI) methods, MDI approaches, and MDI-embedded MI approaches. We found that the MDI approach outperformed its competitors regarding bias and mean squared error in small sample sizes through extensive simulation.

**Keywords:** complete-case analysis; imputation; missing by design; substitution



## 1. Introduction

We consider situations where the covariates of interest are only observable within a detection interval, referred to as the limit of detection (LOD) problem, e.g., [1,2]. The limit of detection is commonly encountered in observational studies. For example, patients with a positive real-time polymerase chain reaction (PCR) test result for coronavirus disease usually indicate that the viral RNA load is greater than the lower LOD, which varies from $10^2$ to $10^6$ copies per milliliter [3]. On the other hand, a typical droplet digital PCR test is restricted to an upper LOD or interval LOD [4]. Assays with high lower LODs or low higher LODs will likely result in higher false-negative or false-positive rates, respectively [3]. Another example where incomplete data due to the LOD are inevitable is in multi-omics data analysis, where quantitative omics measurements, such as metabolite levels and protein expressions, are missing due to the failure of the measurement assay at levels outside of its detection limits [5]. Proper adjustments are needed for valid data analysis with missing values due to the LOD.

In the presence of LOD, one of the most straightforward approaches is the complete-case (CC) analysis, which discards observations that fall outside the detection limits. Despite the CC analysis yielding unbiased estimates for the regression coefficients, e.g., [1,2,6–8], it could suffer from efficiency loss and become unstable when the sample size is small or when there are multiple covariates subject to the LOD. An approach that does not require discarding observations is to substitute the unobserved values with fixed values outside of the detection limits [9]. Common substitution methods replace missing values with ad hoc fix values or values derived from parametric assumptions, e.g., [7,8,10]. Such substitution methods could result in considerable bias when the imputed value is very different from the unobserved values or when the parametric assumptions are misspecified [1,2,7,10,11]. An approach to handle covariates subject to the LOD without discarding observations nor imposing a parametric distributional assumption is the missing indicator (MDI) approach,

e.g., [6,10,12,13]. The idea of the MDI approach is to create a binary variable that indicates whether the covariate of interest is observed and include the indicator variable in the model as an additional covariate [6]. Since the MDI approach uses all of the available observations, estimating procedures that utilize the MDI approach is expected to yield a more efficient estimator and be more computationally stable when the sample size is small or multiple covariates are subject to LOD [14].

Approaches for LOD have been well studied in the literature. For example, the MDI approach was justified theoretically and numerically and compared to the CC approach in the context of linear regression [6] and logistic regression [15–17]. Extensions that combine the MDI and the multiple imputation (MI) approaches have also been studied under the generalized linear regression setting [18,19]. On a separate note, MDI-based and MI-based copula models were used to estimate the association between two continuous variables subjected to lower LOD [10]. Relatively fewer works compared approaches for LOD with survival outcomes. Among those, most of the existing works focus on proportional hazards models, e.g., [20,21]. Despite having a more favorable interpretability, the approaches for LOD under the accelerated failure time (AFT) framework were less explored until recently [22], where a seminonparametric distribution is recommended to model the error term.

Recent studies on the MDI approach yield encouraging results when the required conditions are met. However, most existing works are based on scenarios in which the covariates of interest are subjected to a lower LOD. We extended the MDI approach's applicability to general scenarios in which covariates may be subjected to upper or interval LOD. We applied the MDI approach to the context of survival analysis when the survival time is subject to an independent right censoring. We assumed the survival time is related to the covariates via a parametric accelerated failure time model. We compared the performance of the MDI approach to that of existing methods at different types of LOD via large-scale simulation studies. We compared the approaches by evaluating the absolute value of the average biases (AAB) and mean squared error (MSE) for the regression parameter of interest.

The rest of the paper is organized as follows. Notation and model formulation are presented in Section 2. A brief description of the estimating procedures in the presence of covariates subject to the LOD are provided in Section 3. Results of large-scale simulation studies on the performance of the proposed estimators are reported in Section 4. A discussion concludes Section 5.

## 2. Notations and Model

Let $T_i$ be the time to an event of interest related to covariates via a parametric AFT model,

$$\log(T_i) = \alpha + X_i^{*\top}\beta + Z_i^{\top}\gamma + \epsilon_i, i = 1,\ldots,n, \tag{1}$$

where $X_i^*$ is a $p \times 1$ covariate vector whose elements are partially missing due to LOD, $Z_i$ is a $q \times 1$ fully observed covariate vector, $\epsilon_i$s are independent and identically distributed random variables with a known distribution, and $(\alpha, \beta, \gamma)$ are the corresponding conformable regression parameters. In the absence of the missing covariate $X_i^*$, the regression coefficients can be estimated via maximizing the likelihood function. For example, when $\epsilon_i$ has a normal distribution with mean zero and variance $\sigma^2$, $T_i$ follows a log-normal distribution and the regression coefficients can be obtained by maximizing the likelihood

$$L(\alpha, \gamma, \sigma; \Theta) = \prod_{i=1}^{n} \left[ \frac{1}{\sigma} \phi \left( \frac{\log(Y_i) - \alpha - Z_i^{\top}\gamma}{\sigma} \right) \right]^{\Delta_i} \left[ \Phi \left( -\frac{\log(Y_i) - \alpha - Z_i^{\top}\gamma}{\sigma} \right) \right]^{1-\Delta_i},$$

where $Y_i = \min(T_i, C_i)$ is the observed survival time, $\Delta_i = I(T_i \leq C_i)$ is the censoring indicator, and $C_i$ is the censoring time. The functions $\phi(\cdot)$ and $\Phi(\cdot)$ are the probability density function and the cumulative distribution function of the standard normal distribution. The maximum likelihood estimator (MLE) can be obtained by a standard numerical optimiza-

tion algorithm, such as the Newton–Raphson method. The variance–covariance matrix of the MLE can be estimated via the information matrix, and the asymptotic normality of the MLE follows directly from likelihood theorems. The `survreg()` function from `R`'s `survival` package [23] is available for fitting such a parametric AFT model.

When $X_i^*$ is subject to LOD, we assume $X_{ij}^*$ is observable only if $L_j \leq X_{ij}^* \leq U_j$, where $L_j$ and $U_j$ are the lower and upper bounds of the measurement range, respectively. When $X_{ij}^*$ falls outside of $[L_j, U_j]$, we observe $X_{ij} = \max\{L_j, \min(X_{ij}^*, U_j)\}, j = 1, \ldots, p$. That is, we observe $X_{ij} = L_j$ if $X_{ij}^* < L_j$ and $X_{ij} = U_j$ if $X_{ij}^* > U_j$ so that the direction of missing is always known. Accompanying $X_i = (X_{i1}, \ldots, X_{ip})^\top$ is the missing indicator $V_i = (V_{i1}, \ldots, V_{ip})^\top$, where $V_{ij} = I(L_j \leq X_{ij}^* \leq U_j)$ and $I(\cdot)$ is the indicator function. The observed data then consist of independent copies of $\Theta = \{Y_i, \Delta_i, X_i, Z_i, V_i, L_1, \ldots, L_p, U_1, \ldots, U_p\}, i = 1, \ldots, n$. We assume the censoring time $C_i$ is conditionally independent of $T_i$ given $X_i$ and $Z_i$. Throughout the manuscript, we allow $X_{ij}^*$ to be subject to different types and levels of LOD and discuss approaches that are applicable under these scenarios.

## 3. Estimating Procedures in the Presence of LOD

### 3.1. Complete-Case Analysis

The CC analysis is commonly used in the presence of missing covariates. The fundamental idea of applying the CC analysis is to discard missing observations outside of the measurement range. Though the idea of the CC approach is straightforward, discarding observations from samples loses information and could potentially bias the estimation when the missingness is dependent on exposure, e.g., [16,24], as in the LOD cases. Additional convergence issues arise when the original sample size is small; an extreme case is where the CC approach is inapplicable when all subjects have at least one missing variable. With the missing indicator, the CC model can be expressed as a modification of (1) as follows:

$$Q_i \log(T_i) = Q_i \alpha_c + Q_i X_i^\top \beta_c + Q_i Z_i^\top \gamma_c, i = 1, \ldots, n, \tag{2}$$

where $Q_i = \prod_j V_{ij} = 1$ if all of $X_{ij}, j = 1, \ldots, p$, are observed, and is zero otherwise. The regression coefficients $\{\alpha_c, \beta_c, \gamma_c\}$ can be obtained by maximizing the modified likelihood,

$$\prod_{i=1}^n \left[ \frac{1}{\sigma} \phi \left( \frac{\log(Y_i) - \alpha_c - X_i^\top \beta_c - Z_i^\top \gamma_c}{\sigma} \right) \right]^{\Delta_i Q_i} \left[ \Phi \left( -\frac{\log(Y_i) - \alpha_c - X_i^\top \beta_c - Z_i^\top \gamma_c}{\sigma} \right) \right]^{(1 - \Delta_i) Q_i}.$$

The CC method is the default approach in `survreg()` when data contain missing values.

### 3.2. Parametric Substitution Approaches

Instead of discarding missing observations, imputations methods replace the missing values with their expectations. Most existing imputation methods replace missing values with the predicted values from models trained by the observed data, resulting in imputed values inside the observable region. Such imputation methods are not feasible for imputing missing values due to LOD, where the missing values are outside of the observable region. For this reason, it is more appropriate to consider imputation methods that replace missing values subject to LOD with conditional expectations, $E(X_{ij}^*|X_{ij}^* < L_j)$ or $E(X_{ij}^*|X_{ij}^* > U_j)$, respectively, depending on the direction of missing. These quantities can be estimated parametrically using likelihood methods. For example, for a positive $X_{ij}^*$ subject to a lower LOD by $L_j > 0$, common substitution values such as $L_j/2$ and $L_j/\sqrt{2}$ are derived by imposing a uniform distribution or a triangular distribution to the data below $L_j$, respectively, e.g., [25,26]. On the other hand, ad hoc substituting values such as 0 and $L_j$ have also been considered but generally lead to biased estimations of regression coefficient estimates [9].

Although the aforementioned substituting values have simple forms, they are derived without using information from the observed $X_{ij}^*$. An alternative approach is to derive the substituting values by imposing a distribution assumption on the whole data. For example,

if $X_{ij}^*$ is assumed to follow a normal distribution with mean $\mu_j$ and variance $\varsigma_j^2$, then $(\mu_j, \varsigma_j^2)$ can be estimated by maximizing the likelihood

$$\prod_{i=1}^{n} \left[ \frac{1}{\varsigma} \phi\left( \frac{x_{ij} - \mu_j}{\varsigma_j} \right) \right]^{V_{ij}} \left[ \Phi\left( \frac{L_j - \mu_j}{\varsigma_j} \right) \right]^{I(V_{ij}=0, X_{ij}^*<L_i)} \left[ \Phi\left( \frac{\mu_j - U_j}{\varsigma_j} \right) \right]^{I(V_{ij}=0, X_{ij}^*>U_i)}.$$

Let $r(x) = \phi(x)/\Phi(x)$, and $\widehat{\mu}_j$ and $\widehat{\varsigma}_j^2$ be the MLEs of $\mu_j$ and $\varsigma_j^2$, respectively. Once $\widehat{\mu}_j$ and $\widehat{\varsigma}_j^2$ are obtained, the conditional expectations

$$E(X_{ij}^*|X_{ij}^* < L_j) = \widehat{\mu}_j - \widehat{\varsigma}_j r\left( \frac{L_j - \widehat{\mu}_j}{\widehat{\varsigma}_j} \right) \text{ and } E(X_{ij}^*|X_{ij}^* > U_j) = \widehat{\mu}_j + \widehat{\varsigma}_j r\left( \frac{\widehat{\mu}_j - U_j}{\widehat{\varsigma}_j} \right),$$

can be used as the substituting values for those $X_{ij}^*$ censored by $L_j$ and $U_j$, respectively. The estimates of the regression coefficients in (1) under the parametric substitution methods are then obtained by maximizing the likelihood in (1) with missing $X_{ij}^*$ replaced by the desired substituting values.

### 3.3. Parametric Multiple Imputation Approaches

Single imputation methods such as those mentioned in Section 3.2 are less computation-demanding compared to the MI [27] approaches, but the latter could be more efficient as they better reflect uncertainty about imputed values. The general idea of MI methods is to impute the missing $X_{ij}^*$ repeatedly with values generated from its predictive distribution given the observed data. Once the $M$ complete data sets are generated, the CC analysis is then applied to each complete data set. The separate results are then pooled to provide the final inference. Building onto the aforementioned substitution method under normal assumptions, we consider imputing the missing $X_{ij}^*$s by random values generated from densities $f(x|X_{ij}^* < L_j, \widehat{\mu}_j, \widehat{\sigma}_j^2)$ or $f(x|X_{ij}^* > U_j, \widehat{\mu}_j, \widehat{\sigma}_j^2)$. Under the normal assumption on $X_{ij}^*$, $f(x|\cdot)$ corresponds to truncated normal density functions and the random values are generated via the inverse cumulative distribution function method. Let $\widehat{\theta}_m = (\widehat{\alpha}_m, \widehat{\beta}_m, \widehat{\gamma}_m), m = 1, \dots, M$, be the coefficient estimate obtained by maximizing (1) at the $m$th imputation. Using the Rubin's rule [27], the pooled MI coefficient estimate and variance estimate are

$$\widehat{\theta}_{MI} = \frac{1}{M} \sum_{m=1}^{M} \widehat{\theta}_m \text{ and } Var(\widehat{\theta}_{MI}) = \frac{1}{M} \sum_{m=1}^{M} Var(\widehat{\theta}_m) + \left( 1 + \frac{1}{M} \right) \frac{\sum_{m=1}^{M} (\widehat{\theta}_m - \widehat{\theta}_{MI})^2}{M - 1},$$

where $Var(\widehat{\theta}_n)$ is the variance estimate for $\widehat{\theta}_m$. The proposed MI method differs from the existing MI methods, such as the ones implemented in `mice` [28], in that the proposed method targets imputation values outside of the observed region. Our MI method can be easily implemented and is flexible in that different parametric assumptions can be implied for different covariates.

### 3.4. Missing Indicator Approaches

A useful alternative that does not require discarding or imputing missing values is the MDI approach [6]. The idea of the MDI approach is to include the missing status as additional covariates in the model so that all available information remains in the analysis to maintain statistical power. Specifically, we consider the MDI-embedded AFT model

$$\log(T_i) = \alpha_m + (V_i \circ X_i)^\top \beta_m + Z_i^\top \gamma_m + (1 - V_i)^\top \theta_m + \epsilon_i, i = 1, \dots, n, \tag{3}$$

where $u \circ v$ is the element-wise product of vectors $u$ and $v$ and $\theta_m$ is an additional $p \times 1$ regression coefficient. The MLE of $(\alpha_m, \beta_m, \gamma_m, \theta_m)$ can be obtained by maximizing the modified likelihood

$$\prod_{i=1}^{n} \left[ \frac{1}{\sigma} \phi \left\{ \frac{e_i(\alpha_m, \beta_m, \gamma_m, \theta_m)}{\sigma} \right\} \right]^{\Delta_i} \left[ \Phi \left\{ -\frac{e_i(\alpha_m, \beta_m, \gamma_m, \theta_m)}{\sigma} \right\} \right]^{1-\Delta_i}, \tag{4}$$

where $e_i(\alpha_m, \beta_m, \gamma_m, \theta_m) = \log(Y_i) - \alpha_m - (V_i \circ X_i)^\top \beta_m - Z_i^\top \gamma_m - (1 - V_i)^\top \theta_m$. In the context of linear regression, the least-squares estimator for $\beta_m$ was shown to be asymptotically unbiased for $\beta$ in (1) if $X_i^*$ and $Z_i$ are uncorrelated [6]. The performance of the MDI approach has also been studied under the generalized linear model, e.g., [15]. Since the parametric AFT model has a log-linear form, the MLE obtained from maximizing (4) is expected to be asymptotically unbiased in the absence of censoring. We also conjecture that the asymptotic unbiasedness continue to hold in the presence of censoring. The MDI approach is easy to implement and can be extended in several directions. For example, the fully expanded MDI model extends (3) by including interaction terms between the missing indicators and the observed covariates [15], resulting in the revised AFT model

$$\log(T_i) = \alpha_m + (V_i \circ X_i)^\top \beta_m + Z_i^\top \gamma_m + (1 - V_i)^\top \theta_m + [(1 - V_i) \circ Z_i]^\top \phi_m + \epsilon_i,$$

where $\phi_m$ is an additional $q \times 1$ regression coefficient. On the other hand, the MDI approach could be embedded into the MI approach, e.g., [18,19], resulting in the revised AFT model

$$\log(T_i) = \alpha_m + \widetilde{X}_i^\top \beta_m + Z_i^\top \gamma_m + (1 - V_i)^\top \theta_m + \epsilon_i, i = 1, \dots, n,$$

where $\widetilde{X}_i = (\widetilde{X}_{i1}, \dots, \widetilde{X}_{ip})^\top$, $\widetilde{X}_{ij} = X_{ij}^*$ if $V_{ij} = 1$, and $\widetilde{X}_{ij}$ is the imputed value by MI if $V_{ij} = 0$. The MI coefficient estimates are then pooled by the Rubin's rule. Those extensions of the MDI approach are implemented and compared in simulation.

## 4. Simulation

A series of simulation studies were conducted to compare methods discussed in Section 3. The failure time $T_i$ was generated from the AFT model

$$\log(T_i) = \beta_0 + \beta_1 X_{i1}^* + \beta_2 X_{i2}^* + \gamma_1 Z_i + \epsilon, \tag{5}$$

where $X_{i1}^*$ was a Weibull random variable with shape 1 and scale 1/3, $X_{i2}^*$ was a normal random variable with mean 0 and variance 0.64, $Z_i$ was a standard normal random variable, the regression parameter $(\beta_0, \beta_1, \beta_2, \gamma_1) = (-2, 1, -1, 1)$, and the error term $\epsilon$ followed a standard normal distribution. We considered scenarios where covariates are independent and where the covariates are correlated. In the latter case, the Clayton copula with a Spearman's rho of 0.4 was used to specify the correlation between $X_1$ and $Z$. The censoring time was independently generated from a uniform distribution over $[0, 1.25]$, yielding a 30% censoring rate on $T_i$. We considered three types of LOD: lower LOD, upper LOD, and interval LOD, where $X_{ij}^*$ is observable in $[L_j, \infty]$, $[-\infty, U_j]$, and $[L_j, U_j]$, respectively. The detection limits, $L_j$ and $U_j$, were quantiles of $X_{ij}^*$ chosen to achieve three levels of missing proportions, 20%, 40%, and 60%, for light missing, moderate missing, and heavy missing, respectively. For interval LOD, we additionally assumed $L_j$ to be the $(100 \cdot m_j/4)$th quantile of $X_{ij}^*$, where $m_j$ is the missing proportion for $X_{ij}^*$, $j = 1, 2$.

For each configuration, we compared the performance of the following approaches to handling missing data.

**Complete-case analysis**

    **M1** removal of subjects with missing $X_{ij}^*$.

**Substitution methods:**

    **M2** substitution of the missing $X_{ij}^*$ by $L_j/2$ or $2U_j$.

    **M3** substitution of the missing $X_{ij}^*$ by $L_j/\sqrt{2}$ or $\sqrt{2}U_j$.

    **M4** substitution of the missing $X_{ij}^*$ by $E(X_{ij}^*|X_{ij}^* < L_j)$ or $E(X_{ij}^*|X_{ij}^* > U_j)$ under normal assumptions.

**Multiple imputation approaches:**

**M5** MI of the missing $X_{ij}^*$ using the predictive mean matching (PMM) algorithm implemented in the `R` package `mice` [28].

**M6** MI of the missing $X_{ij}^*$ using conditional densities derived under normal assumptions as described in Section 3.3.

**Missing indicator approaches:**

**M7** the missing indicator approaches (MDI) model.

**M8** the expanded MDI model.

**Missing-indicator-embedded multiple imputation approaches (MI + MDI):**

**M9** MI by PMM and fit with MDI model.

**M10** MI by normal assumptions and fit with MDI model.

**M11** MI by PMM and fit with expanded MDI model.

**M12** MI by normal assumptions and fit with expanded MDI model.

The simulation was repeated 10,000 times with sample sizes $n = 50, 100$, and 500. The MLE of the regression parameter of the AFT model (5) was obtained using the `survreg()` function in the `survival` package [23] in `R` [29] under the normal error assumption, e.g., with argument `dist = "lognormal"`. For the scenarios considered, the CC approach (M1) sometimes failed to converge as the resultant sample size was too small or empty after removing missing observations. The convergence rate for the CC approach under different scenarios presented in the Supplementary Material shows fewer converged replications when the sample size is small (e.g., $n = 50$) or the missing proportions are high (e.g., $m_1 = 60\%$ or $m_2 = 60\%$). For this reason, the simulation results were based on the converged replications for the CC approach. For MI methods, the number of imputations $M$ was set to 5.

Tables 1 and 2 summarize the AAB and MSE associated with the MLEs of $\beta_1$, $\beta_2$, and $\gamma_1$ in the AFT model (5) when the covariates are independent and the censored covariates are subjected to a lower LOD. The MDI approaches (M7 and M8) have among the smallest AAB and MSE across the considered scenarios. Moreover, the MDI approaches outperform the CC approach (M1) when the sample size is small or the missing proportions ($m_1$ and $m_2$) are high. Overall, the AAB and the MSE generally increase with increasing missing proportions. On the other hand, whereas MSE generally decreases with an increasing sample size, the trend of AAB varies by model. Among the substitution methods, both M2 and M3 yield smaller AAB for $\beta_1$ than for $\beta_2$; this is because the substituting values under these approaches are close to $E(X_{i1}^*|X_{i1}^* < L_1)$. On the contrary, M4 yields smaller AAB for $\beta_2$ when the parametric assumption for $X_2$ is satisfied. The same trend can be seen in the parametric MI approach, M6. In particular, all of the imputation approaches, including the PMM-based MI approach (M5), did not improve the performance when compared with the MDI approach. Combining MDI models in MI approaches does not necessarily improve the performance of MDI or MI approaches if they would be applied solely. In situations where the combined approach shows improved AAB over the MI approaches, there are trade-offs in MSE. Of those, the expanded MDI-embedded MI approach (M11 and M12) yields smaller AAB than the MDI-embedded MI approach (M9 and M10), but they result in a comparable MSE. In addition, biases associated with the MLEs of $\beta_1$ and $\beta_2$ summarized in Figure 1 provide insight into the direction of bias. Among those that yield a substantial bias, approaches with uniform and triangular assumptions, i.e., M2 and M3, tend to overestimate $\beta_1$ and underestimate $\beta_2$. In contrast, approaches with normal assumptions, i.e., M4, M6, and M10, tend to underestimate $\beta_1$ and correctly estimate $\beta_2$. The pattern is reversed in the case of an upper or interval LOD. These observations suggest that the direction of bias is imposed by the underlying parametric assumption and highlight the robustness of the MDI approach. Similar trends are observed in scenarios where the covariates are subjected to the upper or interval LOD and where n = 500, as presented in the Supplementary Materials. On the other hand, the results when the covariates are correlated are presented in Tables 3 and 4 and Figure 2. For all approaches,

correlation generally results in higher AAB and MSE but does not alter the direction of bias. This observation is consistent with the literature, where the asymptotic bias of the regression coefficient associated with the censored covariate is shown to increase with an increasing magnitude of the correlation [6]. However, these theoretical results do not apply directly to a small sample setting, as the MDI approaches remain at least as good as, if not better than, the CC approach.

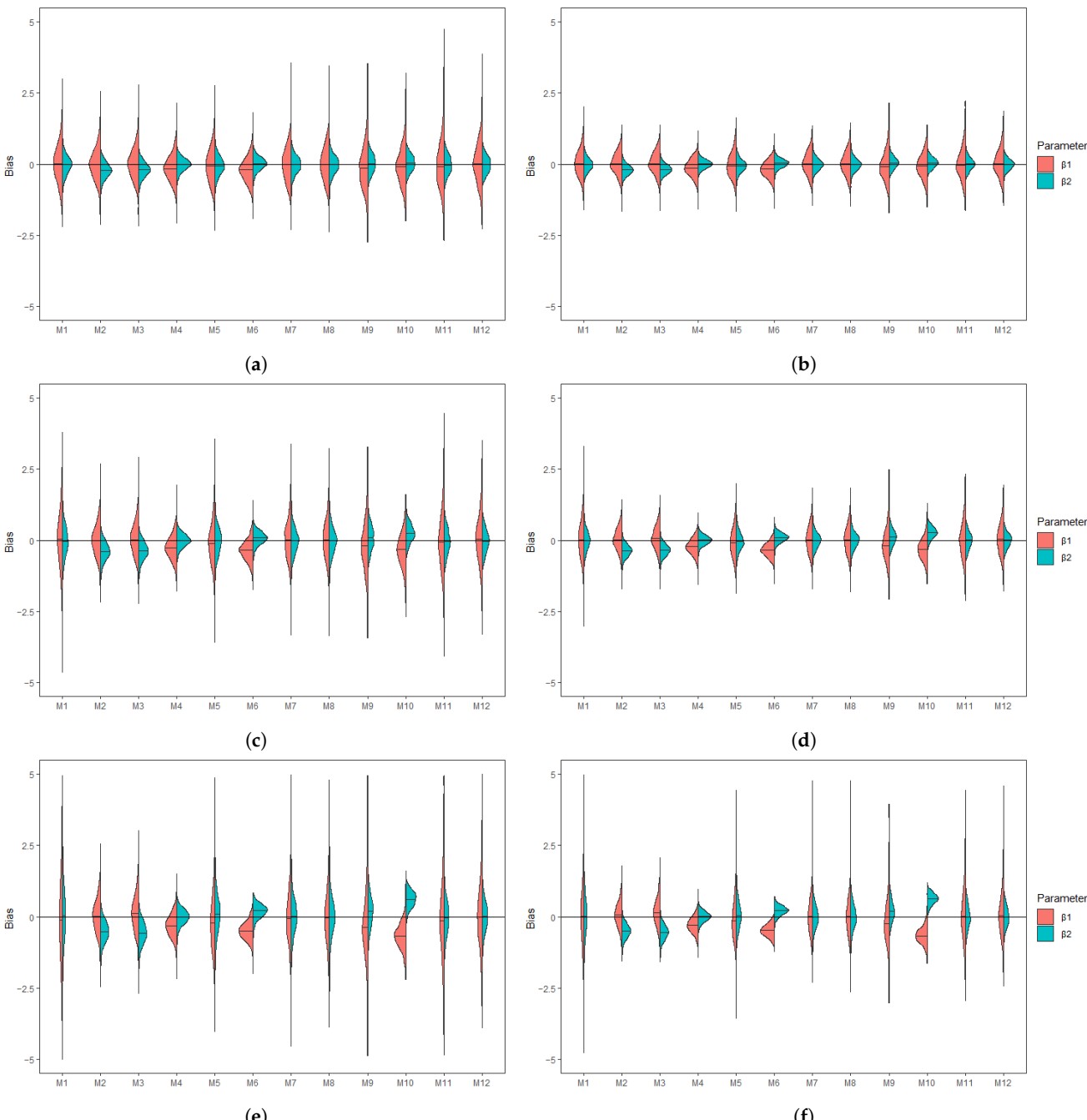

**Figure 1.** Violin plots showing the empirical distribution of the bias associated with MLE of $\beta_1$ (red) and $\beta_2$ (green) when covariates are independent and $X_{ij}^*, j = 1, 2$ is subjected to lower LOD. (**a**) Bias under $n = 50$ and $m_1 = m_2 = 20\%$. (**b**) Bias under $n = 100$ and $m_1 = m_2 = 20\%$. (**c**) Bias under $n = 50$ and $m_1 = m_2 = 40\%$. (**d**) Bias under $n = 100$ and $m_1 = m_2 = 40\%$. (**e**) Bias under $n = 50$ and $m_1 = m_2 = 60\%$. (**f**) Bias under $n = 100$ and $m_1 = m_2 = 60\%$.

**Table 1.** Summary of the AAB ($\times 1000$) when covariates are independent and $X_{ij}^*, j = 1, 2$ is subjected to lower LOD. M1 is complete-case analysis; M2–M4 are the different variants of the substitution methods; M5–M6 are the different variants of the MI methods; M7–M8 are the different variants of the MDI methods; M9–M12 are the different variants of MDI-embedded MI (MI + MDI) methods. AAB less than 0.1 is highlighted in gray, with darker tones corresponding to smaller AAB.

| $n$ | | Substitution | | | | MI | | MDI | | MI + MDI | | | |
|---|---|---|---|---|---|---|---|---|---|---|---|---|---|
| | | **M1** | **M2** | **M3** | **M4** | **M5** | **M6** | **M7** | **M8** | **M9** | **M10** | **M11** | **M12** |
| | | | | | | $m_1 = m_2 = 20\%$ | | | | | | | |
| 50 | $\beta_1$ | 20 | 7 | 7 | 149 | 54 | 193 | 6 | 11 | 98 | 68 | 34 | 36 |
| | $\beta_2$ | 32 | 219 | 198 | 9 | 55 | 15 | 19 | 20 | 20 | 34 | 18 | 24 |
| | $\gamma_1$ | 2 | 10 | 10 | 5 | 15 | 5 | 4 | 2 | 33 | 1 | 41 | 2 |
| | | | | | | $m_1 = m_2 = 40\%$ | | | | | | | |
| | $\beta_1$ | 80 | 1 | 43 | 248 | 43 | 345 | 39 | 42 | 136 | 302 | 11 | 73 |
| | $\beta_2$ | 28 | 393 | 373 | 9 | 23 | 73 | 20 | 20 | 92 | 241 | 34 | 24 |
| | $\gamma_1$ | 5 | 11 | 11 | 8 | 23 | 6 | 8 | 10 | 41 | 6 | 65 | 1 |
| | | | | | | $m_1 = m_2 = 60\%$ | | | | | | | |
| | $\beta_1$ | 40 | 38 | 142 | 298 | 74 | 489 | 39 | 46 | 226 | 671 | 29 | 77 |
| | $\beta_2$ | 213 | 540 | 584 | 12 | 102 | 190 | 5 | 2 | 213 | 592 | 44 | 5 |
| | $\gamma_1$ | 191 | 12 | 12 | 12 | 61 | 9 | 14 | 45 | 103 | 5 | 155 | 0 |
| | | | | | | $m_1 = m_2 = 20\%$ | | | | | | | |
| 100 | $\beta_1$ | 27 | 28 | 38 | 124 | 16 | 169 | 31 | 28 | 52 | 60 | 15 | 41 |
| | $\beta_2$ | 15 | 200 | 180 | 6 | 51 | 30 | 12 | 11 | 51 | 52 | 10 | 8 |
| | $\gamma_1$ | 7 | 5 | 5 | 7 | 3 | 7 | 8 | 4 | 39 | 17 | 45 | 14 |
| | | | | | | $m_1 = m_2 = 40\%$ | | | | | | | |
| | $\beta_1$ | 43 | 36 | 76 | 219 | 53 | 328 | 35 | 30 | 133 | 316 | 36 | 43 |
| | $\beta_2$ | 4 | 365 | 347 | 8 | 30 | 100 | 9 | 6 | 115 | 271 | 2 | 2 |
| | $\gamma_1$ | 7 | 4 | 4 | 5 | 1 | 7 | 5 | 7 | 58 | 17 | 79 | 10 |
| | | | | | | $m_1 = m_2 = 60\%$ | | | | | | | |
| | $\beta_1$ | 70 | 65 | 161 | 282 | 68 | 477 | 69 | 63 | 165 | 670 | 83 | 88 |
| | $\beta_2$ | 35 | 504 | 545 | 9 | 66 | 220 | 5 | 8 | 226 | 629 | 3 | 11 |
| | $\gamma_1$ | 22 | 3 | 3 | 5 | 8 | 8 | 4 | 18 | 78 | 15 | 118 | 8 |

**Table 2.** Summary of the MSE ($\times 1000$) when covariates are independent and $X_{ij}^*, j = 1, 2$ is subjected to lower LOD. M1 is complete-case analysis; M2–M4 are the different variants of the substitution methods; M5–M6 are the different variants of the MI methods; M7–M8 are the different variants of the MDI methods; M9–M12 are the different variants of MDI-embedded MI (MI + MDI) methods. MSEs less than 0.1 are highlighted in gray, with darker tones corresponding to smaller MSEs.

| $n$ | | Substitution | | | | MI | | MDI | | MI + MDI | | | |
|---|---|---|---|---|---|---|---|---|---|---|---|---|---|
| | | **M1** | **M2** | **M3** | M4 | **M5** | **M6** | **M7** | **M8** | **M9** | **M10** | **M11** | **M12** |
| | | | | | | $m_1 = m_2 = 20\%$ | | | | | | | |
| 50 | $\beta_1$ | 530 | 366 | 372 | 288 | 500 | 280 | 476 | 485 | 640 | 447 | 764 | 562 |
| | $\beta_2$ | 126 | 131 | 115 | 52 | 111 | 53 | 98 | 101 | 131 | 102 | 142 | 111 |
| | $\gamma_1$ | 45 | 33 | 33 | 31 | 38 | 32 | 32 | 43 | 139 | 69 | 148 | 73 |
| | | | | | | $m_1 = m_2 = 40\%$ | | | | | | | |
| | $\beta_1$ | 1376 | 393 | 433 | 285 | 768 | 299 | 699 | 736 | 795 | 451 | 1099 | 793 |
| | $\beta_2$ | 457 | 276 | 254 | 58 | 231 | 61 | 186 | 194 | 230 | 189 | 255 | 206 |
| | $\gamma_1$ | 107 | 35 | 34 | 33 | 46 | 34 | 35 | 82 | 249 | 56 | 269 | 58 |
| | | | | | | $m_1 = m_2 = 60\%$ | | | | | | | |
| | $\beta_1$ | >9999 | 445 | 585 | 305 | 1585 | 368 | 1418 | 1559 | 1616 | 675 | 2526 | 1547 |
| | $\beta_2$ | >9999 | 470 | 538 | 78 | 531 | 99 | 412 | 445 | 493 | 459 | 550 | 448 |
| | $\gamma_1$ | >9999 | 36 | 36 | 36 | 60 | 36 | 38 | 193 | 513 | 52 | 579 | 50 |

**Table 2.** *Cont.*

| n | | Substitution | | | | MI | | MDI | | MI + MDI | | | |
|---|---|---|---|---|---|---|---|---|---|---|---|---|---|
| | | **M1** | **M2** | **M3** | **M4** | **M5** | **M6** | **M7** | **M8** | **M9** | **M10** | **M11** | **M12** |
| | | | | | | $m_1 = m_2 = 20\%$ | | | | | | | |
| 100 | $\beta_1$ | 239 | 164 | 165 | 128 | 239 | 132 | 204 | 207 | 272 | 188 | 314 | 234 |
| | $\beta_2$ | 56 | 79 | 68 | 24 | 52 | 25 | 45 | 45 | 58 | 45 | 59 | 46 |
| | $\gamma_1$ | 21 | 14 | 14 | 14 | 16 | 14 | 14 | 20 | 57 | 27 | 60 | 27 |
| | | | | | | $m_1 = m_2 = 40\%$ | | | | | | | |
| | $\beta_1$ | 501 | 172 | 192 | 142 | 360 | 179 | 300 | 301 | 352 | 243 | 452 | 318 |
| | $\beta_2$ | 147 | 188 | 172 | 28 | 99 | 37 | 76 | 76 | 107 | 126 | 102 | 77 |
| | $\gamma_1$ | 42 | 14 | 14 | 14 | 19 | 14 | 14 | 34 | 93 | 22 | 100 | 21 |
| | | | | | | $m_1 = m_2 = 60\%$ | | | | | | | |
| | $\beta_1$ | 2069 | 202 | 273 | 168 | 663 | 279 | 608 | 617 | 566 | 533 | 850 | 646 |
| | $\beta_2$ | 749 | 330 | 379 | 35 | 251 | 78 | 161 | 166 | 262 | 439 | 224 | 169 |
| | $\gamma_1$ | 142 | 15 | 15 | 15 | 23 | 16 | 15 | 66 | 191 | 20 | 209 | 18 |

**Table 3.** Summary of the AAB ($\times 1000$) when covariates are correlated and $X_{ij}^*, j = 1, 2$ is subjected to lower LOD. M1 is complete case analysis; M2–M4 are the different variants of the substitution methods; M5–M6 are the different variants of the MI methods; M7–M8 are the different variants of the MDI methods; M9–M12 are the different variants of MDI-embedded MI (MI + MDI) methods. AAB less than 0.1 is highlighted in gray, with darker tones corresponding to smaller AAB.

| n | | Substitution | | | | MI | | MDI | | MI + MDI | | | |
|---|---|---|---|---|---|---|---|---|---|---|---|---|---|
| | | **M1** | **M2** | **M3** | **M4** | **M5** | **M6** | **M7** | **M8** | **M9** | **M10** | **M11** | **M12** |
| | | | | | | $m_1 = m_2 = 20\%$ | | | | | | | |
| 50 | $\beta_1$ | 52 | 29 | 41 | 127 | 7 | 188 | 42 | 43 | 16 | 52 | 14 | 74 |
| | $\beta_2$ | 27 | 188 | 169 | 11 | 23 | 36 | 14 | 16 | 54 | 74 | 18 | 22 |
| | $\gamma_1$ | 5 | 8 | 5 | 17 | 31 | 8 | 2 | 9 | 65 | 2 | 57 | 5 |
| | | | | | | $m_1 = m_2 = 40\%$ | | | | | | | |
| | $\beta_1$ | 114 | 29 | 77 | 244 | 0 | 366 | 102 | 111 | 47 | 300 | 64 | 139 |
| | $\beta_2$ | 26 | 354 | 336 | 16 | 15 | 102 | 11 | 13 | 128 | 270 | 1 | 16 |
| | $\gamma_1$ | 4 | 1 | 3 | 7 | 44 | 13 | 7 | 7 | 90 | 13 | 72 | 22 |
| | | | | | | $m_1 = m_2 = 60\%$ | | | | | | | |
| | $\beta_1$ | 39 | 70 | 189 | 296 | 17 | 511 | 235 | 390 | 112 | 640 | 109 | 283 |
| | $\beta_2$ | 105 | 497 | 540 | 17 | 129 | 223 | 30 | 39 | 249 | 612 | 6 | 40 |
| | $\gamma_1$ | 25 | 15 | 18 | 10 | 25 | 37 | 18 | 46 | 105 | 31 | 106 | 27 |
| | | | | | | $m_1 = m_2 = 20\%$ | | | | | | | |
| 100 | $\beta_1$ | 21 | 22 | 31 | 133 | 32 | 193 | 24 | 21 | 71 | 83 | 21 | 37 |
| | $\beta_2$ | 6 | 186 | 167 | 15 | 32 | 41 | 9 | 9 | 68 | 66 | 30 | 8 |
| | $\gamma_1$ | 4 | 0 | 2 | 12 | 44 | 3 | 5 | 6 | 57 | 0 | 41 | 11 |
| | | | | | | $m_1 = m_2 = 40\%$ | | | | | | | |
| | $\beta_1$ | 20 | 25 | 65 | 234 | 75 | 359 | 25 | 23 | 138 | 338 | 9 | 40 |
| | $\beta_2$ | 7 | 353 | 335 | 20 | 13 | 109 | 5 | 2 | 115 | 273 | 5 | 5 |
| | $\gamma_1$ | 13 | 8 | 11 | 1 | 66 | 17 | 11 | 17 | 68 | 14 | 52 | 20 |
| | | | | | | $m_1 = m_2 = 60\%$ | | | | | | | |
| | $\beta_1$ | 168 | 44 | 143 | 307 | 33 | 515 | 86 | 84 | 142 | 668 | 43 | 103 |
| | $\beta_2$ | 23 | 493 | 534 | 18 | 51 | 223 | 9 | 11 | 191 | 605 | 26 | 9 |
| | $\gamma_1$ | 25 | 23 | 27 | 19 | 72 | 48 | 26 | 30 | 76 | 39 | 77 | 35 |

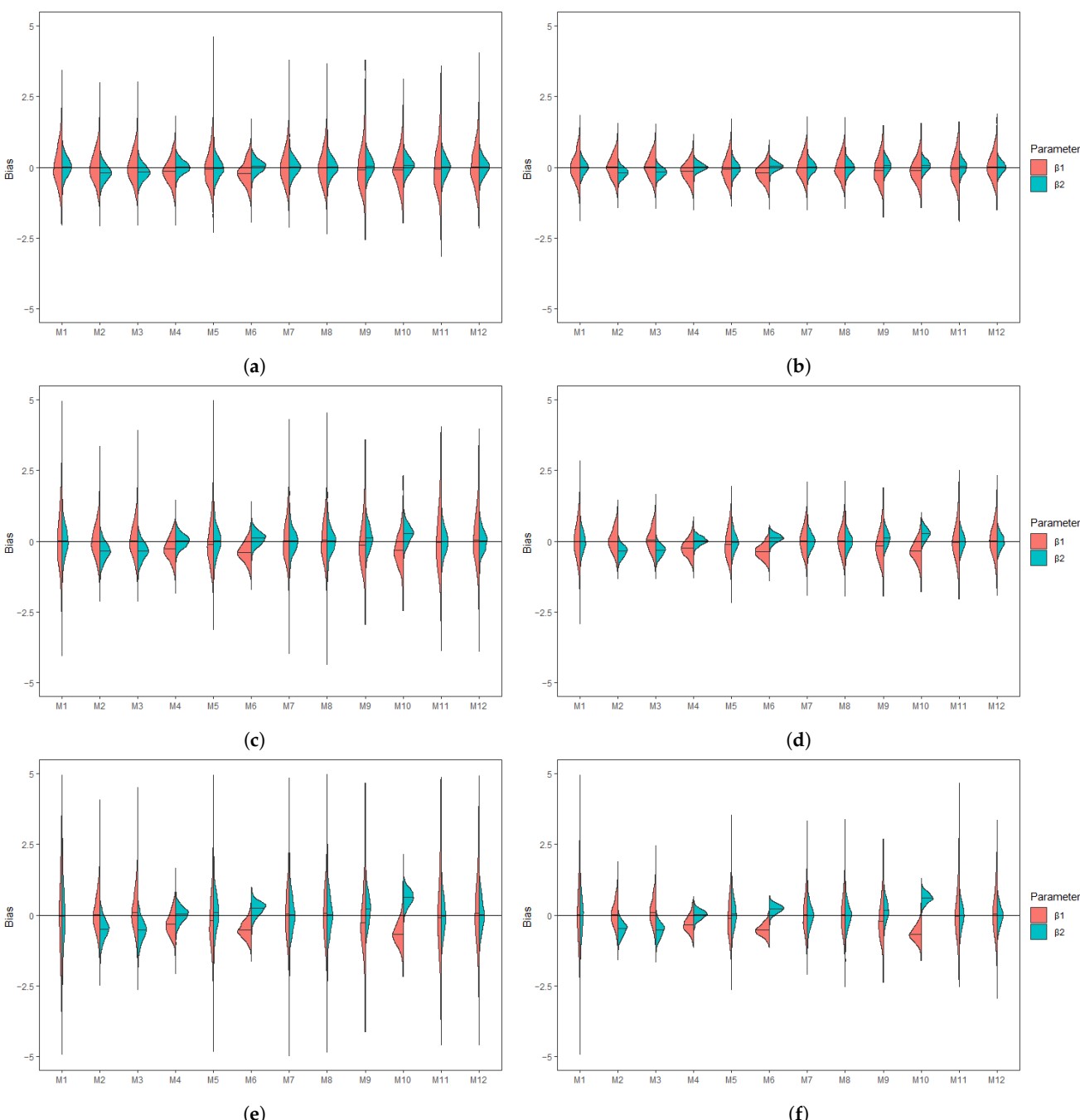

**Figure 2.** Violin plots showing the empirical distribution of the bias associated with MLE of $\beta_1$ (red) and $\beta_2$ (green) when covariates are correlated and $X_{ij}^*, j = 1, 2$ is subjected to lower LOD. (**a**) Bias under $n = 50$ and $m_1 = m_2 = 20\%$. (**b**) Bias under $n = 100$ and $m_1 = m_2 = 20\%$. (**c**) Bias under $n = 50$ and $m_1 = m_2 = 40\%$. (**d**) Bias under $n = 100$ and $m_1 = m_2 = 40\%$. (**e**) Bias under $n = 50$ and $m_1 = m_2 = 60\%$. (**f**) Bias under $n = 100$ and $m_1 = m_2 = 60\%$.

**Table 4.** Summary of the MSE ($\times 1000$) when covariates are correlated and $X_{ij}^*, j = 1, 2$ is subjected to lower LOD. M1 is complete case analysis; M2–M4 are the different variants of the substitution methods; M5–M6 are the different variants of the MI methods; M7–M8 are the different variants of the MDI methods; M9–M12 are the different variants of MDI-embedded MI (MI + MDI) methods. MSEs less than 0.1 are highlighted in gray, with darker tones corresponding to smaller MSEs.

| | | Substitution | | | | MI | | MDI | | MI + MDI | | | |
|---|---|---|---|---|---|---|---|---|---|---|---|---|---|
| $n$ | | M1 | M2 | M3 | M4 | M5 | M6 | M7 | M8 | M9 | M10 | M11 | M12 |
| | | | | | | $m_1 = m_2 = 20\%$ | | | | | | | |
| 50 | $\beta_1$ | 613 | 395 | 397 | 273 | 565 | 264 | 519 | 543 | 642 | 432 | 798 | 598 |
| | $\beta_2$ | 128 | 118 | 104 | 52 | 112 | 53 | 96 | 99 | 132 | 103 | 134 | 108 |
| | $\gamma_1$ | 61 | 35 | 34 | 34 | 40 | 33 | 36 | 57 | 143 | 66 | 170 | 69 |
| | | | | | | $m_1 = m_2 = 40\%$ | | | | | | | |
| | $\beta_1$ | 1742 | 424 | 482 | 272 | 962 | 294 | 901 | 933 | 864 | 455 | 1203 | 997 |
| | $\beta_2$ | 405 | 244 | 225 | 60 | 231 | 68 | 188 | 191 | 250 | 206 | 263 | 204 |
| | $\gamma_1$ | 155 | 35 | 35 | 34 | 46 | 34 | 38 | 107 | 236 | 52 | 275 | 54 |
| | | | | | | $m_1 = m_2 = 60\%$ | | | | | | | |
| | $\beta_1$ | >9999 | 556 | 789 | 316 | 2760 | 384 | 3314 | >9999 | 2158 | 625 | 3449 | 3606 |
| | $\beta_2$ | >9999 | 420 | 481 | 80 | 520 | 116 | 456 | 488 | 552 | 491 | 638 | 489 |
| | $\gamma_1$ | 1539 | 37 | 37 | 36 | 53 | 37 | 38 | 255 | 513 | 49 | 601 | 48 |
| | | | | | | $m_1 = m_2 = 20\%$ | | | | | | | |
| 100 | $\beta_1$ | 274 | 192 | 195 | 159 | 237 | 160 | 225 | 226 | 265 | 195 | 305 | 245 |
| | $\beta_2$ | 55 | 68 | 58 | 21 | 48 | 23 | 42 | 43 | 64 | 45 | 63 | 44 |
| | $\gamma_1$ | 29 | 16 | 16 | 16 | 19 | 16 | 17 | 28 | 66 | 28 | 70 | 29 |
| | | | | | | $m_1 = m_2 = 40\%$ | | | | | | | |
| | $\beta_1$ | 550 | 200 | 221 | 166 | 359 | 212 | 320 | 327 | 340 | 263 | 429 | 347 |
| | $\beta_2$ | 143 | 173 | 157 | 23 | 100 | 35 | 75 | 77 | 108 | 127 | 105 | 79 |
| | $\gamma_1$ | 60 | 16 | 16 | 16 | 24 | 16 | 16 | 45 | 102 | 21 | 112 | 21 |
| | | | | | | $m_1 = m_2 = 60\%$ | | | | | | | |
| | $\beta_1$ | 2590 | 226 | 302 | 195 | 688 | 323 | 632 | 647 | 552 | 525 | 797 | 668 |
| | $\beta_2$ | 694 | 312 | 362 | 31 | 231 | 76 | 157 | 159 | 237 | 411 | 216 | 162 |
| | $\gamma_1$ | 207 | 17 | 17 | 16 | 29 | 18 | 17 | 89 | 188 | 22 | 203 | 20 |

## 5. Discussion

The MDI approach minimizes the loss of information and does not require making parametric assumptions, making it an attractive alternative to some of the more widely used approaches for handling missing covariates. Moreover, the MDI approaches show clear advantages over the competitors and are recommended in models with survival outcomes, as in our simulation. Our simulation shows no apparent difference between the MDI and the expanded MDI models, but embedding the expanded MDI model in MI could result in a higher bias reduction. The advantage of the MDI approach is more substantial when there is a large proportion of missing covariates or when the distributional assumption is violated in the MI approach. The MDI approaches continue to perform well under additional simulation settings, including scenarios where the survival time is not subject to censoring and scenarios under a Cox proportional hazard model setting.

It has been noted that, even though the MDI approach generally results in a reduced bias, it might have minimal improvements when the missing mechanism is associated with the outcome [30] or when the missing covariate is categorical [31]. Those phenomena were verified in the context of generalized linear regression, and it would be worth investigating those scenarios in our setting with survival outcomes. Moreover, extending the assessments of the validity of the MDI approach, e.g., [32,33], to our settings will be of interest.

We only considered scenarios where the direction of missing is known in this paper. Nevertheless, the MDI approach is still applicable when the direction of missing is unknown. The aforementioned parametric imputation methods can easily be extended to the case when the direction of missing is unknown. For example, suppose that $X_{ij}^*$ follows a normal

distribution with mean $\mu_j$ and variance $\varsigma_j^2$ as in Section 3.3. The MLEs of $\mu_j$ and $\varsigma_j^2$ can be obtained by maximizing the likelihood

$$\prod_{i=1}^{n} \left[ \frac{1}{\varsigma} \phi \left( \frac{x_{ij} - \mu_j}{\varsigma_j} \right) \right]^{V_{ij}} \left[ \Phi \left( \frac{L_j - \mu_j}{\varsigma_j} \right) + \Phi \left( \frac{\mu_j - U_j}{\varsigma_j} \right) \right]^{1-V_{ij}}.$$

The corresponding MI procedure can then be carried out with missing $X_{ij}^*$s imputed by values generated from density $pf(x|X_{ij}^* < L_j, \widehat{\mu}_j, \widehat{\sigma}_j^2) + (1-p)f(x|X_{ij}^* > U_j, \widehat{\mu}_j, \widehat{\sigma}_j^2)$, where $p = 1$ with probability $\Phi[(L_j - \widehat{\mu}_j)/\widehat{\varsigma}_j]/\{\Phi[(L_j - \widehat{\mu}_j)/\widehat{\varsigma}_j] + \Phi[(\widehat{\mu}_j - U_j)/\widehat{\varsigma}_j]\}$ and $p = 0$ otherwise. Due to its simplicity, the MDI method can also be easily embedded into other methods to improve the overall performance. An immediate example is the MI+MDI approaches considered in Section 4. Another extension is to embed the MDI approach in threshold regression approaches [34] to accommodate multiple censored covariates.

**Supplementary Materials:** The following supporting information can be downloaded at: https://www.mdpi.com/article/10.3390/stats5020029/s1.

**Author Contributions:** Conceptualization, N.A. and S.H.C.; methodology, N.A. and S.H.C.; software, N.A.; validation,N.A. and S.H.C.; formal analysis, N.A. and S.H.C.; writing—original draft preparation, N.A.; writing—review and editing, S.H.C.; visualization, N.A. and S.H.C.; supervision, S.H.C.All authors have read and agreed to the published version of the manuscript.

**Funding:** This research received no external funding.

**Institutional Review Board Statement:** Not applicable.

**Informed Consent Statement:** Not applicable.

**Conflicts of Interest:** The authors declare no conflict of interest.

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
