# Peer review of "The Missing Indicator Approach for Accelerated Failure Time Model with Covariates Subject to Limits of Detection"

_stats, doi:10.3390/stats5020029_

Round 1

Reviewer 1 Report

I enjoyed reading this manuscript because of how the authors defined every concept and symbol clearly. This makes the write accessible to readers interested in understand missing data mechanisms. I believe the simulations are adequate, as are the conclusions based on the simulations. Some minor edits are needed in the writing, but overall it's an outstanding manuscript.

Author Response

We appreciate the encouraging comments. We have carefully proofread the manuscript again in this revision to ensure no lingering mistakes.

Reviewer 2 Report

This manuscript conducted a series of survival analysis simulations to compare the performance of twelve handling missing data approaches under different settings of limits of detection, derived from the complete-case analysis, substitution methods, multiple imputation methods, and missing indicator methods, and missing indicator embedded multiple imputation methods, and found that both missing indicator approaches perform well regarding the metrics of absolute value of average biases and mean squared error. The manuscript is well done. These results will help further studies dealing with missing data subjected to the limit of detection.

Author Response

We appreciate the encouraging comments. In this revision, we have carefully proofread the manuscript again to ensure no lingering mistakes and included additional simulations to consolidate our conclusions.

Reviewer 3 Report

This paper utilises a simulation study to investigate different approaches to handle limit of detection within survival analysis data.

Specific comments regarding the manuscript are given below.

Comments:

  1. The two sample sizes utilised within the simulation study are both quite small. It would good for the analysis to be undertaken with another larger sample size to showcase how the methods behave with more data. Do the differences in the approaches hold when sufficient data is available? Or are the recommendations made within the manuscript only for small sample analyses?
  2. It is mentioned within the manuscript that the CC approach suffers from convergence issues due to small sample sizes. As the results provided are based on the converged replications, the number of converged replications under different scenarios needs to be provided.
  3. Focus is given to the absolute bias when comparing the different approaches. Was the direction of the bias also investigated?
  4. The plots within Figure 1 are difficult to follow. What do the red and green areas represent – is this for the two parameters? More explanation is needed within the text and a legend should be added to the figure.
  5. As the results/findings of the paper focus solely around a simulation study, the syntax used for this should be included within supplementary material.
  6. The paper would have benefited from analysis of real-life data in addition to the simulation study to showcase the real-world consequences (if any) of utilising the different approaches.

Minor suggestions/typos:

  1. Ensure all terminology is formally defined.
  2. Is there a typo within the description of Tables 1 & 2? Should this say “darker tones” rather than “darker tunes”?

Reviewer 4 Report

The Authors compared the effectiveness of different alternatives to the complete case (CC) approach under different Limit of Detection (LOD) settings for AFT survial model. They considered multiple imputation (MI) methods, missing indicator (MDI) approaches.
The Author concluded based on simulation study that the MDI method outperformed its competitors regarding bias and mean squared error.

The paper is interesting and discusses many of the imputation methods used in the (LOD) case. 
Some drawback of the paper is the lack of new methods and theoretical results.  Only one small size AFT model is considered. 
Maybe it is worth to consider more such models, apply to Cox proportional hazards model, consider situations of correlated predictors.

Round 2

Reviewer 3 Report

Thank you for providing a detailed response to the original comments. All comments have been taken into account in the updated manuscript.

Reviewer 4 Report

In the current version of the work, the authors took my suggestions into account.
I am satisfied with the new version of the work.